# The Role of the Kinase Inhibitors in Thyroid Cancers

**DOI:** 10.3390/pharmaceutics14051040

**Published:** 2022-05-11

**Authors:** Francesca Cuomo, Claudio Giani, Gilda Cobellis

**Affiliations:** 1Department of Experimental Medicine, Università degli Studi della Campania L. Vanvitelli, 80138 Napoli, Italy; francesca.cuomo@unicampania.it; 2School of Medicine, University of Pisa, 56124 Pisa, Italy; claudio.giani48@gmail.com; 3Department of Precision Medicine, Università degli Studi della Campania L. Vanvitelli, 80138 Napoli, Italy

**Keywords:** cancer, thyroid, TKI therapy, chemoresistance, trials

## Abstract

Thyroid cancer is the most common endocrine malignancy, accounting for about 3% of all cancer cases each year worldwide with increasing incidence, but with the mortality remaining stable at low levels. This contradiction is due to overdiagnosis of indolent neoplasms identified by neck ultrasound screening that would remain otherwise asymptomatic. Differentiated thyroid carcinomas (DTCs) are almost curable for 95% with a good prognosis. However, 5% of these tumours worsened toward aggressive forms: large tumours with extravasal invasion, either with regional lymph node or distant metastasis, that represent a serious clinical challenge. The unveiling of the genomic landscape of these tumours shows that the most frequent mutations occur in tyrosine kinase receptors (RET), in components of the MAPK/PI3K signalling pathway (RAS and BRAF) or chromosomal rearrangements (RET/PTC and NTRK hybrids); thus, tyrosine-kinase inhibitor (TKI) treatments arose in the last decade as the most effective therapeutic option for these aggressive tumours to mitigate the MAPK/PI3K activation. In this review, we summarize the variants of malignant thyroid cancers, the molecular mechanisms and factors known to contribute to thyroid cell plasticity and the approved drugs in the clinical trials and those under investigation, providing an overview of available treatments toward a genome-driven oncology, the only opportunity to beat cancer eventually through tailoring the therapy to individual genetic alterations. However, radiotherapeutic and chemotherapeutic resistances to these anticancer treatments are common and, wherever possible, we discuss these issues.

## 1. Introduction

### 1.1. Differentiated Thyroid Cancer

Differentiated thyroid cancers (DTCs) derive from follicular cells, the major cell type of thyroid gland deputed to thyroid hormones production and secretion, representing >90% of all TCs and including three different histologic types of neoplasia: papillary (PTC), follicular (FTC) and Hurthle cell carcinoma (HTC) [1]. PTC is the most common, accounting for 80–85% for total TC in countries with inadequate iodine intake.

The overall survival (OS) of DTC at 5 years is very high (98%). Despite this, local recurrence (thyroid lodge o neck lymph nodes) and/or distant metastases are detected in 20% and 10% of patients, respectively [2]. Generally, the prognosis of PTC is better than FTC. The less favourable prognosis of FTC may be related to the patients’ older age and to the extent of the tumour, in particular the vascular invasion. Furthermore, the prognosis may differ for the respective histologic or cytologic variants: in PTC, poor prognoses are represented by the tall cell, the columnar cell and the oxyphilic variants, whereas in FTC by HTC and insular lesions. These DTC variants represent a group of cancers that loses the papillary nuclear features with increased numbers of mitoses and necrosis with elevated risk of local relapse, persistence and metastases, and higher mortality. These variants of DTC are generally indicated as poorly differentiated thyroid cancer (PDTC) that progressively lose the capacity to take up iodine by metastatic cells, decreasing the OS rate (less than 10% at 10 years), and a further important prognostic factor is the stage at diagnosis (TNM): for stage I and II, the OS is included between 75 and 95% and fall to 60% to <50% for stage 3 or 4, respectively [3].

#### Treatment of DTC

Generally, the treatment of differentiated thyroid cancer includes surgery and eventually the postsurgical thyroid ablation with radioiodine. The extension of thyroid surgery depends at least in part on the dimension of the nodule. For patients with suspicious cancer <1 cm, clinically limited on the lobe, with no evidence of extrathyroidal extension or node metastatic disease or prior neck irradiation, thyroid lobo-isthmectomy is an adequate initial surgical treatment. Total thyroidectomy is indicated with at least one of the following anatomic or clinical conditions: suspicious thyroid nodule >4 cm or unilateral o multifocal disease associated with clinically o intraoperative detected cervical nodes metastases or metastatic disease to distant sites or gross and evident extrathyroidal extension [4]. Both total surgical or lobo-isthmectomy may proposed to patients with suspicious nodule >1 and <4 cm with no evidence of extrathyroidal extension, lymph nodes or distant metastases. It is important to point out that only total thyroidectomy enables the ablation with radioiodine and consequent improvement of follow-up accuracy. The indication should be evaluated on the risk of mortality and of disease recurrence or persistence (AJCC/UICC staging VIII Edition) and the risk stratification system (ATA 2018). In patients with ATA low/risk (T1a-b, No Mo), radioiodine ablation is not generally recommended; in patients with intermediate or low/intermediate risk remnant, ablation should be considered in particular in advanced age, in patients with large tumours, macroscopic lymph nodes or aggressive histology and/or cytology and/or vascular invasion. In the patients with high risk or intermediate/high risk, radio-ablation is recommended. In general, the indication for the post-surgical thyroid ablation with radioiodine should also be evaluated on post-surgical levels of Thyroglobulin (TG) and neck ultrasound; in the presence of the basal or TSH-stimulated TG undetectable or <5 ng/mL associated with negative neck ultrasound (US), the probability of local or metastatic disease is negligible and the radio-ablation of residual thyroid tissue can be avoided [4]. In one third of advanced differentiated thyroid carcinoma, the metastatic tissue loses the ability to uptake radioiodine with a consequent lack of efficacy of radioiodine therapy, and pronounced decrease in OS rate (less than 10% at 10 years) (RAI-refractory DTC). In these tumours with local/neck growth, the local regional beam radiation (ERBT) should be considered for palliation of locoregional recurrences or in the presence of metastatic bone lesions for the control of local pain. For patients with RAI-refractory DTC or with metastatic DTC disease that progresses despite standard therapy, significantly symptomatic and/or with the life threatening no suitable for local therapies, systemic therapy should be planned. Traditional chemotherapy with Doxorubicin remains the single effective and approved cytotoxic therapy; however, it is not recommended often as first-line treatment [5].

### 1.2. Poorly Differentiated and Anaplastic Thyroid Cancer

Poorly differentiated thyroid cancer (PDTC) and anaplastic thyroid cancer (ATC) are rare and represent about 3% of all thyroid cancers. They are the most aggressive forms that show resistance to traditional multidisciplinary therapies such as surgery, radiation and chemotherapy. In fact, the median survival of patients with such diagnoses is only 4–6 months. Association between surgical resection of tumoral tissue present in the neck, chemotherapy with doxorubicin and cisplatin associated with locoregional radiotherapy showed limited efficacy [4,6]. The progression of PDTC and ATC is very rapid, especially at a loco-regional neck level, with invasion of tissues including the trachea, muscles, oesophagus and larynx. Distant metastases can be found in the lungs, bone, skin and brain [6].

### 1.3. Medullary Thyroid Cancer

Medullary thyroid cancer (MTC) arises from parafollicular or C Cells of the thyroid and accounts for 5% of all thyroid cancers [7]. The clinical and biological behaviour of MTC is more severe than that of PTC: in fact, the 10-year survival rate is about 50%. MTC can be sporadic or hereditary. Sporadic MTC can clinically appear at any age, with the highest incidence during the fifth and sixth decades of life. Hereditary MTC represents 20 to 30% of all MTCs with an autosomal dominant pattern of transmission and high penetrance. It can be transmitted as a single entity, familial MTC, or as a part of multiple endocrine neoplasia (MEN) syndromes type IIa or Iib. Distant metastases are observed at presentation in 7–23% of MTC patients and are the main cause of MTC-related deaths. For localized MTC, total thyroidectomy and central compartment neck dissection represent the only curative treatment. In the case of persistence or relapse of local regional disease, EBRT is indicated to improve the local control of disease. In addition, EBRT may be used for the control of locoregional metastases or as palliative therapy to reduce pain for bone metastases or to reduce brain metastases. Radiofrequency thermo-ablation is frequently applicable to treat single bone, liver and lung metastases. Another promising local treatment is transarterial chemoembolization (TACE) generally used in cases of liver metastases, when lesions are smaller than 3 cm and the liver involvement less than 30%. In diffuse metastatic disease, chemotherapy demonstrated no clinically durable advantages or benefits. Recently, promising results have been demonstrated for peptide receptor radionuclide therapy using 90Y-labeled somatostatin analogues; this therapeutic approach is limited to MTC with significant somatostatin receptors expression [8].

## 2. Signalling Pathways Activated in Thyroid Cancers

Cancer genomics data derived from different hystotypes of thyroid cancers (Cancer Genome Atlas, Cosmic and cBioportal) have identified frequent mutations in receptor tyrosine kinases, in components of the mitogen-activated protein kinase (MAPK) and phosphatidylinositol 3-kinase (PI3K) pathways, chromosomal rearrangements and in tumour suppressor gene p53, or PTEN and TERT genes (Figure 1) [9,10].

Receptor tyrosine kinases (RTKs) are cell surface receptors for growth factors, hormones, cytokines, key regulators of normal cellular processes, sharing a common architecture: the extracellular ligand binding domain and the intracellular tyrosine kinase domain [11]. Upon ligand binding to the extracellular domain and subsequent receptor dimerization, the intracellular domain catalyses its autophosphorylation, propagating the signal through the tyrosine phosphorylation of several different substrates to downstream transduction pathways, mainly MAPK cascade.

At the molecular level, when any effector of these pathways is mutated, the signalling became constitutively activated, leading to uncontrolled proliferation and enhanced cell survival that initiate the transformation of follicular thyroid cells to PTC and FTC. Among these, nearly 70% of PTCs carry point mutations in RAS or BRAF protooncogenes or RET gene (Figure 1).

RET encodes a tyrosine kinase receptor activated by GDNF family ligands. After the ligand binding, RET dimerizes and activates the RAS/MAPK and PI3K/AKT pathways that control organ development and tissue homeostasis. In 1985, the RET gene was discovered as the most frequent mutated gene in thyroid cancer. Three general mechanisms lead to aberrant RET activation: in-frame RET fusions, targeted mutation and aberrant overexpression of the RET gene [12,13]. Rearrangements of RET are associated with thyroid cancer, such as RET/PTC1 and RET/PTC3, which are produced by fusion of RET tyrosine-kinase domain to the 5′ terminal region of coiled-coil domain containing gene 6 (CCDC6) [14] and RET/PTC3 with Ele1 [15], found in children affected by PTC after the Chernobyl accident in 1986.

Mutations affecting the extracellular cysteine-rich domain of RET (C634 mainly associated with MEN2A) result in covalent dimerization and, constitutively, activation of the receptor [16]. Mutations affecting the intracellular domain of RET, M918T, usually associated with FMTC, sporadic MTC and always with MEN2B, signal independently of GDNF apparently as monomeric oncoproteins [17].

Other less frequent chromosomal rearrangements affecting other receptor genes such as NTRK (1.2%), ALK (0.8%) and FGFR2 (0.4%) are found in PTCs with the highest prevalence of NTRK fusions in radiation-exposed populations [5].

RAS proteins, encoded by three ubiquitously expressed genes HRAS, KRAS and NRAS, are GTPases able to activate the MAPK signalling pathways (Figure 1). Eight percent of DTC, 17% of FTC, 21% of PDTC and 18% of ATC carry a mutation in codon 12, 13 or 61 of NRAS genes [10]. When RAS is mutated, it promotes the constitutive activation of MAPK signalling that contributes uncontrolled proliferation. The similar mutation frequency in different histotypes may suggest that RAS is responsible for thyroid cancer initiation, but not for progression. It is noteworthy to highlight that RAS activation induces dedifferentiation in a dose-dependent manner, inhibiting thyroid-specific gene expression [18].

The BRAF gene encodes for a serine-threonine kinase that signals from cell membrane to the nucleus, activating cell growth (Figure 1). Discovered in 1982, the oncogene is an important driver in thyroid carcinogenesis being sufficient for tumour initiation and progression, leading to progressive loss of differentiation markers, such as Tireoglobulin (Tg) and sodium-iodide symporter (NIS) controlling the iodine uptake gradually lost and responsible for iodine refractoriness [19,20]. The most frequent mutation is BRAF^V600E^, a substitution from valine (V) to glutamic acid (E) in codon 600 of exon 15, found in 99% of all thyroid cancers, in the 58.6% of DTC with higher prevalence in the tall cells variant, in 33% of PDTC and in 45% of ATC [10,21].

Less frequent mutations have been found in AKT and PI3KCA, the PI3K kinase effectors controlling cell survival. The high frequency of PI3K/AKT mutations in ATC opens the possibility that these genes are mutated late in the progression, when the tumour is already established [22].

PTEN is a protein and lipid phosphatase able to promote dephosphorylation of PIP3, switching off the PI3K signalling. PTEN is a tumour suppressor gene, and the defective protein allows the cell to divide in an uncontrolled way and prevents damaged cells from death, leading to the growth of tumours. More than 70 mutations have been identified in the PTEN gene and 3–10% of patients develop thyroid cancer. Fifteen percent of ATC carries a PTEN mutation [10].

TERT encodes for the telomerase reverse transcriptase, controlling telomere length linked to PTC aggressiveness. The C228T and C250T mutations are present in nearly 10% of PTC, in 15% of FTC, in 42% of PDTC and in 73% of ATC [10,21].

In FTC and FVPTC, Pax8/PPARg fusions covering 30% of cases have been identified, highlighting the importance also for genes involved in thyroid organogenesis [21]. It is important to note that the hybrid Pax8/PPARg has not be found in ATC so far.

Single-cell transcriptome analyses depicted a more detailed genomic landscape of thyroid cancers, underlying PTC initiation and progression, identifying a premalignant thyrocyte subpopulation that evolves in different malignant states [23].

When a second mutation occurs, the loss of differentiation promotes the evolution toward PDTC. Epigenetic alterations, p53 or TERT mutations might be late mutations in the evolution of ATC, the most lethal cancer [21].

Mutation hierarchy in thyroid cancers is nicely evidenced by data generated for the most aggressive thyroid cancers, PDTC and ATC, carrying mutations in TERT promoter gene (42.3% in PTDC and 73.5% in ATC), suggesting that the initial BRAF+ve subpopulation is lost during the tumour progression and the TERT+ve cells have a positive growth selection over the others, becoming more aggressive [23].

## 3. The TKI Approved Drugs in Thyroid Cancers

Activation of the pathways described led to the development of many small molecules, called Tyrosine Kinase Inhibitors (TKIs) able to inhibit tyrosine kinases and/or tyrosine kinases receptors (TKR), and several clinical studies demonstrated their efficacy in both solid and non-solid tumours. Nowadays, TKI have become ground-breaking, purpose-designed treatments that hold the potential to greatly improve outcomes for thyroid cancer patients.

The sequence and structure similarities among the kinase domains of these molecules lead to the generation of inhibitors able to block contemporary multiple targets. In addition, TKIs are cytostatic, but not cytotoxic and have a potent antiangiogenic activity with possible consequent shrinkages of tumour lesions.

To date, FDA and EMA approved different TKIs as the first line treatment either in advanced or metastatic RAI-refractory DTCs [24]. When disease progression occurs even during TKI therapy, the possibility to shift to another TKI is indicated based on evidence that the second drug has higher possibility of being effective, hoping that the inhibition of activated signalling pathway is superior.

The first TKI drugs approved for advanced differentiated thyroid cancers are Sorafenib and Lenvatinib.

Sorafenib is an oral TKI, found as BRAF inhibitor and later as pan-kinase inhibitor, blocking VEGFR1/3, PDGF, FGF, KIT and RET mutations, with a strong antiangiogenic activity. The phase 3 clinical trial DECISION led to the approval of Sorafenib for treatment of iodine-refractory DTC [25]. The primary endpoint of the study showed that progression-free survival was 10.8 months in the Sorafenib group compared to 5.8 months in the placebo group (*p* < 0.001) with manageable toxicity, leading to frequent hand-foot syndrome, diarrhoea and hypertension as adverse effects (AEs).

Lower efficacy was seen in few patients with ATC in a phase II trial, but Sorafenib did not show superiority to fosbretabulin treatment, a microtubule-destabilising drug, the lead compound in a class of agents termed vascular disrupting agents (VDAs) [26].

Lenvatinib is a small oral molecule with antiangiogenic activity, inhibiting VEGFR1/3, FGF1/4, PDGF, KIT and RET receptor. The phase III study, SELECT, showed the efficacy of Lenvatinib in naïve TKI patients with DTC and MTC, with a PFS of 18.3 months compared to 3.6 months in the placebo group. Patients that were already receiving a TKI treatment before the Lenvatinib showed a longer PFS in this subgroup and the overall survival increased in patients older than 65 [27].

Lenvatinib was also evaluated in ATC, but although fewer benefits were demonstrated, it has not been approved for ATC treatment [28,29].

The efficacy of Lenvatinib has been also corroborated by quality-of-life (QOL) studies demonstrating the efficacy of this treatment [30], positioning it as first-line treatment [31].

The similarity of RET with kinase domains led to the approval of TKIs for MTC. Among them, Vandetanib (VEGFR, EGFR and RET inhibitor) and Cabozantinib (MET and VEGFR inhibitor) were first approved for MTC [32].

The ZETA study demonstrated the longer PFS in symptomatic, unresectable, locally or advanced MTC patients treated with Vandetanib with respect to placebo (30.5 months compared to 19.3 months in placebo group) [33,34] and also the quality-of life data showed the efficacy of Vandetanib [31].

In the phase 3 study EXAM, Cabozantinib (140 mg) was used in progressive and metastatic RET ^M918T^ MTC and although the significant difference of PFS (11.2 months vs. 4 months, refs. [35,36]), the severe observed adverse effects in advanced MTC patients led to dose reduction or dose discontinuation; thus, Cabozantinib has been indicated as second-line treatment.

The EXAMINER study is evaluating the effect of lower dose of Cabozantinib (60 mg) in MTC, but the results are not published yet.

In these trials, different RET mutations have not always been considered as selective biomarker. Retrospective analysis of ZETA and EXAM studies showed that RET^M918T^ patients have a higher ORR and PFS with Vandetanib and Cabozantinib, respectively [33,34,35,36,37,38].

In non-small cell lung carcinoma (NSCLC), Cabozantinib showed higher efficacy in RET-fusion patients [37], leaving open the hypothesis that an inhibitor may have different efficacy against RET mutation and/or rearrangement.

The major limitation of using TKIs are the significant and sometimes prohibitive adverse effects in many patients, limiting the use or the dose, leading to drug discontinuation or dose reduction. It is widely accepted that the toxicity came from off-target effects.

A failure is seen with Vemurafenib, a specific BRAF kinase inhibitor used in RAI-refractory PTC with V600E mutation that gave serious adverse effects (squamous cell carcinoma, lymphopenia) in a high percentage of patients in phase II study [39]. For this reason, no phase III was approved. Another specific BRAF kinase inhibitor, Dabrafenib, was used for ATC treatment in combination with Trametinib, a MEK inhibitor in a small group of patients, for which not all variables reached the endpoint [40,41].

The long-term inhibition of cancer cell growth by TKIs has manifested several limitations: progressive inhibition of iodine uptake, acquired chemoresistance by positive selection of dedifferentiated stem-like cells, and induction of gatekeeper mutations.

To overcome the above problems, strong efforts have been dedicated to generate highly specific inhibitors during the discovery process of new kinase inhibitors.

In 2020, high potent RET inhibitors, with selective ATP competitive function, have been developed and moved to clinic: Pralsetinib (BLU-667) and Selpercatinib (LOXO-292). In vitro studies of these drugs showed higher capacity to inhibit a wide spectrum of RET alterations. In the ARROW [42] and LIBRETTO 001 [43] phase I/II trials, the superiority and tolerability of these drugs were evaluated in RET-mutant MTCs patients carrying RET activating mutations and rearrangements, showing encouraging results. Selpercatininb showed a durable antitumoral activity, independent on prior TKI treatments and modest adverse effects. The gatekeeper RET mutations, V804M and V804L, responsible for acquired resistance to the TKIs, showed higher sensitivity to Pralsetinib and Selpercatinib, suggesting that the phenotypic drug discovery can be complemented by the designated target-based approach to become more effective.

The combinatorial effect of TKIs plus the standard radiotherapy or combination of different specific inhibitors may decrease the likelihood of a tumour developing therapy resistance.

ASTRA (NCT01843062), an international, phase III, double-blind trial, studied the combinatorial effect of Selumetinib, a specific MEK inhibitor, in high-risk patients with DTC (primary tumor >4 cm; gross extrathyroidal extension outside the thyroid gland; or N1a/N1b disease with ≥1 metastatic lymph node(s) ≥1 cm or ≥5 lymph nodes) plus adjuvant radiotherapy RAI (131-I; 100 mCi/3.7 GBq), compared to placebo. The primary endpoint was complete remission (CR) and the results showed no improvement in CR (CR rate 18 months after RAI) [44].

The MAPK pathway activation resulted in iodine refractoriness and TKI therapy resulted in re-differentiation of radioiodine-refractory thyroid cancers [45,46]. The activation of the receptor tyrosine kinase erbB-3 (HER3) mitigates the MAPK activation achieved by BRAF inhibitors in BRAF^V600E^ mutant thyroid cancers. Combined inhibition of BRAF and HER3 using Vemurafenib and the human monoclonal antibody CDX-3379, respectively, would restore radioiodine (RAI) avidity by potently inhibiting MAPK activation in patients with BRAF-mutant iodine refractory thyroid cancer [47].

In 2017, Sherman et al. showed the combinatorial effect of Sorafenib with temsirolimus, an inhibitor of mTOR pathway, a component of the phosphatidylinositol 3-kinase (PI3K) cell survival pathway, in radioactive iodine-refractory thyroid cancer with encouraging results [48].

A phase II study on metastatic DTC has been approved and is now recruiting patients (NCT01263951) to evaluate the effect of everolimus, an oral mTOR inhibitor, in combination with Sorafenib, but results are not available yet.

Multiple small BRAF kinase inhibitors have been developed, although the benefits of these inhibitors are short-lived [49]. Small molecule-mediated targeted protein degradation has recently emerged as a novel therapeutic strategy in melanoma, which, in our opinion, can be translated to BRAF^V600E^ PTC cancers. These new molecules target the ubiquitin proteasome system (UPS), degrading selectively the BRAF^V600E^, not the wild-type, impairing the cell growth with these thalidomide-based heterobifunctional compounds [50].

Multiple ongoing studies have been approved and the results will better define the sequential systemic therapy for thyroid cancer patients: the agents currently (or formerly) used in second line after Sorafenib or Lenvatinib (i.e., Cabozantinib) will shift to third line, as the approval of new molecules. Table 1 summarizes the ongoing studies registered at clinicaltrial.gov.

## 4. Drug Resistance in Thyroid Cancer

Development of drug resistance is the most challenging problem for thyroid cancers also. There are several mechanisms that limit the efficacy of current treatments; normal tissue toxicity and pharmacokinetic parameters restrict the recommended dosage of each drug and the amount of the drug reaching cancer cells [51]. Cancer cells are initially sensitive to chemotherapeutic drugs, but they develop resistance through DNA mutations (gatekeeper mutations), metabolic changes that induce drug inhibition and degradation and transcriptional silencing induce by epigenetic modifications.

The most frequent metabolic risk in chemoresistance is to increase efflux through the ATP-binding cassette (ABC) transporters. These transmembrane proteins are evolutionary conserved pumps that avoid accumulation of toxins within cells. Three transporters—multidrug resistance protein 1 (MDR1), multidrug resistance-associated protein 1 (MRP1), and breast cancer resistance protein (BCRP)—are implicated to efflux many xenobiotics, including kinase inhibitors from cells [52]. No detailed expression analysis of these genes has been evaluated in different thyroid cancer histotypes. However, ATP-binding cassette (ABC) transporters have been identified as the major determinants of ATC resistance to chemotherapy and these transporters could be the target in tailoring new treatments for ATC patients [52].

Cancer progenitor cells, a group of tumour cells with stem cell characteristics, have the ability of self-renewal, multi-lineage differentiation and tumour formation, persist in patients seemingly in remission and are able to remain stationary or migrate to other sites generating metastases. Expression of epithelial-mesenchymal transition genes such as Snail-1 play an important role in cancer stem cell (CSC) formation and the acquisition of chemoresistance of thyroid cancer cells [53], causing disease relapse at the original tumour site or in distant organs and should be targeted by most effective therapies.

Uncontrolled cell growth of malignant thyroid tumours should be controlled by inhibiting apoptosis or autophagy processes. Thus, efforts in identifying new compounds able to induce cell death should be taken in consideration.

The next step in thyroid cancer therapy would be the use of epigenetic drugs. In all cancers, DNA methylation changes the healthy regulation of gene expression toward a disease pattern by silencing DNA repair genes as early step in cancer initiation. Hypermethylation of these genes leads to a defective DNA repair process and accumulation of DNA damage gives rise to cancer. Reprogramming of DNA methylation using epidrugs would restore the expression of these genes involved in surveillance of the genome [54]. To date, no ‘epidrug’ compounds have been approved for thyroid cancers.

Data on resistance mutations, identified in most TKI targets, suggests the need for new inhibitors to overcome gatekeeper mutations. As discussed above, Pralsetinib and Selpercatinib, the new highly selective RET inhibitors, showed efficacy in patients carrying the gatekeeper RET mutations, V804M and V804L, responsible for drug resistance. Thus, identification of genomic mutations will help in deciding the better therapy, although not being conclusive; recent studies identified in a small set of patients other RET mutations as non-gatekeeper mutations (V738A, Y806C/N AND G810/C/S), which cause less sensitivity to Selpercatinib [55].

The drug development so far has been centred on the inhibition of increased signalling caused by genetic mutations occurring in any of the effectors along a pathway, such as those for BRAF and MEK1, in receptor tyrosine kinases as for RET and NTRK fusions. However, knowing the activated signalling components or the network behind them is not sufficient to predict qualitative patient outcomes in cancer therapy. A quantitative understanding is crucial to tailor the cancer therapy, since bypass signalling is a common mechanism to develop drug resistance to TKI, especially for the RTK-RAS-RAF-MAPK pathway [56].

Thus, drug resistance can be thought as the way of cancer cells to use a relatively diverse, but finite number of strategies to overcome the drug inhibition.

## 5. Conclusions and Future Perspectives

Treatment for differentiated thyroid cancer (DTC) includes complete total thyroidectomy, followed by radioactive iodine (RAI) therapy for metastatic lesions. Aberrant activation of RTKs, MAPK/PI3K pathways led to Tyrosine kinase inhibitor (TKI) treatment as the final treatment option for metastatic lesions, which is incurable with surgery/RAI therapy, improving the prognosis of patients with advanced thyroid cancer with distant metastasis and progressive disease. However, patients can develop severe adverse effects and cancer cells can acquire TKI resistance by different strategies.

Thus, to tackle the resistance mechanism of thyroid cancer cells, new efforts must be dedicated to using combined drugs with different effects, with each drug used at its optimal dose, without intolerable side effects, including epidrugs, developing covalent inhibitors, developing dual-target inhibitors, targeting lysosomes, facilitating the transition to individualized thyroid cancer treatment required to overcome RAI and chemo-resistance, regardless of tumour type and oncogene.

## Figures and Tables

**Figure 1 pharmaceutics-14-01040-f001:**
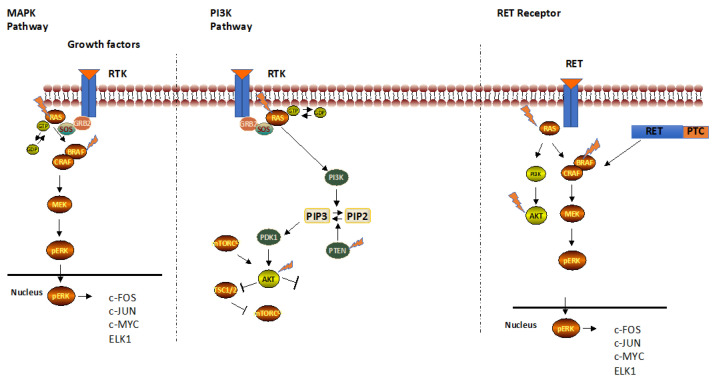
Signalling pathways activated in thyroid cancers leading to uncontrolled thyroid cell growth.

**Table 1 pharmaceutics-14-01040-t001:** Tyrosine Kinase Inhibitors approved in ongoing clinical trials for Thyroid Cancer.

Clinical Trial	Title	Study Start:	Conditions	Interventions	Study Type:	Outcome Measures
NCT05182931	A Prospective, Multi-Centre Trial of TKI Redifferentiation Therapy in Patients with RAIR Thyroid Cancer (I-FIRST Study)	March 2022	Thyroid Cancer	Dabrafenib 75 MG Trametinib 2 MG	Interventional Phase 2	PFS as assessed by RECIST 1.1 criteria at 6 and 12 months in participants who proceed to l131 treatment PFS by RECIST 1.1 criteria at 6 and 12 months in all participants and a control population (SELECT study) Objective response rate by RECIST 1.1 criteria in all treated participants ORR of treated participants
NCT02657551	A Study Using Regorafenib as Second- or Third-Line Therapy in Metastatic MTC	January 2016	Thryoid Cancer	Regorafenib	Interventional Phase 2	PFS Biomarkers associated with response
NCT04521348	Multiple Target Kinase Inhibitor and Anti-Programmed Death-1 Antibody in Patients with Advanced Thyroid Cancer	December 2019	Thryoid Cancer	Multiple tyrosine kinase inhibitor (mTKI) combined with anti-PD-1 antibody	Interventional Phase 2	ORR PFS DOR DCR TTR
NCT04321954	Lenvatinib in Locally Advanced Invasive Thyroid Cancer	March 2021	Differentiated Thyroid Cancer Advanced Cancer	Lenvatinib	Interventional Phase 2	ORR R0/R1 resection rate Number of participants with treatment-related adverse events as assessed by CTCAE v 5.0
NCT04560127	Camrelizumab in Combination with Apatinib in Patients with RAI-refractory Differentiated Thyroid Cancer	September 2020	Radioactive Iodine-refractory Differentiated Thyroid Cancer	Camrelizumab combination with Apatinib	Interventional Phase 2	PFS OR OS DCR TTP AE
NCT04554680	Clinical Trial in RAI-Refractory Thyroid Carcinoma	December 2020	Thryoid Cancer	Dabrafenib and Trametinib	Interventional Phase 2	OR PFS
NCT05007093	Study on the Treatment of DTC with Anlotinib	December 2020	Differentiated Thyroid Cancer	Anlotinib hydrochloride	Interventional Phase 2	TTR PFS
NCT04952493	Anlotinib or Penpulimab in Combination with RAI for DTC	July 2019	Differentiated Thyroid Cancer	Anlotinib +I 131 Penpulimab	Interventional Phase 2	ORR DCR PFS
NCT04238624	Study of Cemiplimab Combined with Dabrafenib and Trametinib in People With ATC	January 2020	Anaplastic Thyroid Cancer with BRAF Gene Mutation	Dabrafenib Trametinib	Interventional Phase 2	ORR PFS OS
NCT04787328	A Study of HA121-28 Tablets in Patients With MTC	July 2021	Medullary Thyroid Carcinoma	HA121-28 tablets	Interventional Phase 2	ORR PFS DOR OS
NCT04544111	PDR001 Combination Therapy for RAI-Refractory Thyroid Cancer	September 2020	Follicular, Poorly Differentiated Carcinoma	Trametinib, Dabrafenib, PDR001	Interventional Phase 2	ORR PFS OS
NCT04061980	Encorafenib and Binimetinib with or Without Nivolumab in Treating Patients with Metastatic RAI-Refractory BRAF Mutant Thyroid Cancer	October 2020	BRAF V600E Mutation Present Refractory Thyroid Carcinoma	Binimetinib Encorafenib Nivolumab	Interventional Phase 2	ORR PFS DOR
NCT05102292	The Efficacy and Safety of HLX208 in Advanced ATC With BRAF V600 Mutation	December 2021	Anaplastic Thyroid Cancer	HLX208	Interventional Phase 1 Phase 2	ORR PFS OR
NCT04739566	Dabrafenib and Trametinib Combination as a Neoadjuvant Strategy in BRAF-positive ATC	January 2021	Anaplastic Thyroid Cancer	Dabrafenib and Trametinib	Interventional Phase 2	ORR OS PFS
NCT04675710	Pembrolizumab, Dabrafenib, and Trametinib Before Surgery for the Treatment of BRAF- Mutated ATC	June 2021	Anaplastic Thyroid Cancer	Dabrafenib Pembrolizumab Trametinib	Interventional Phase 2	ORR OS PFS
NCT04161391	Study of TPX-0046, A RET/SRC Inhibitor in Adult Subjects with Advanced Solid Tumors Harboring RET Fusions or Mutations	December 2019	Medullary Thyroid Cancer	TPX-0046	Interventional Phase 1 Phase 2	ORR AEs
NCT04579757	Surufatinib in Combination with Tislelizumab in Subjects with Advanced Solid Tumors	March 2021	Anaplastic Thyroid Cancer	Surufatinib Tislelizumab	Interventional Phase 1 Phase 2	ORR OS PFS DOR Maximum plasma concentrations of surufatinib and tislelizumab with blood sampling

Abbreviations: Overall response rate (ORR); Overall survival (OS); Progression-free survival (PFS); Duration of Response (DOR); Disease Control Rate (DCR); Time to response (TTR); Time to Progression (TTP); Adverse events (AE); Overall response (OR).

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
