# Peer review of "The Role of the Kinase Inhibitors in Thyroid Cancers"

_pharmaceutics, 2022, doi:10.3390/pharmaceutics14051040_

Round 1

Reviewer 1 Report

 Thyroid cancer is an endocrine malignancy, accounting for about 3% of all cancer cases worldwide. Differentiated thyroid carcinoma (DTC) is the prevalent type of this cancer, derived mainly from follicular cells. Importantly, more than 98% of the cases are curable, but the rest can be developed into aggressive forms that consist a serious clinical challenge. The main driving mutations involved in the onset of the disease occur in the tyrosine kinase receptors RET, c-MET, FGFR or PDGFR as well as components of the AKT (mainly PI3K) or ERK (mainly BRaf and Ras) signaling pathways. In addition, there are also mutations in the tumor suppressors p53 and TERT as well as overexpression of VEGF that may induce tumor vascularization by nearby endothelial cells. Other mutation may exist as well, but those are less abounded. In this review, the authors summarize the variants of malignant thyroid cancers, and molecular mechanisms involved in the cancer initiation and maintenance. Importantly, they also cover the approved drugs in treating the DTC, drugs in clinical trials, and when possible, mechanisms of resistances to these drugs.

Taken together, this review covers an important issue, providing information on DTC and mainly on current clinical trials with various drugs. However, my enthusiasm from this article is diminished due to the similarity to a recent review published elsewhere (Ref. 5). The unique aspects of the current review are not clear.  In addition, the writing of parts of the review are slopy, and most of the sections describing the signaling pathways involved are confusing. These, as well as other comments listed below should be addressed in order to make this review suitable for publication.

Comments

  • The review is similar to the one published recently in Ref 5. The authors should state what is the unique emphasis in the current manuscript.
  • The description of the signaling pathways (section 2, pages 3 and 4) is very confusing. In fact, the authors describe signaling components and not the pathways themselves. They do not describe how is the signals transmitted and what is the role of the tumor suppressor genes involved. The role and regulation of VEGF should be described separately. It is strongly recommended to include a figure that demonstrate the signaling pathways and how they are related to the mutated receptors.
  • It is not clear to me why do the authors use the term TKI, as some of the inhibitors described are in fact inhibitors of Ser/Thr kinases. For example, Sorafenib was in fact developed as a BRaf inhibitor, and was only later shown as a pan kinase inhibitor. Dabrafenib and trametinib are Raf and MEK inhibitors. Although the term TKI might be the common terminology in the thyroid cancer field, it is not accurate, and this point should be mentioned.
  • The drug resistance section is not well explained as it lacks a huge body of literature on the mechanism of resistance to kinase inhibitors (see PMID: 31614114for example). I am aware that the rate of resistance development in DTC is relatively slow and not all mechanisms are relevant, but at least some of the mechanisms do exist in this cancer and should be mentioned.
  • There are quite a few typographical mistakes. For example, in the affiliations, the numbers 1 and 2 appear twice each. The first sentence in the introduction “DTC derive from follicular cells”. What is MKI (appears twice). What is “thyroid bed o neck lymph nodes”. Something is wrong with the authors of the first reference. These any many more small mistakes, as well as the lack of a proper scheme give an impression that the manuscript is slopy and clearly requires a proper proof reading.

Author Response

Thanks for the comments. We have taken the comments on board to improve and clarify the scientific importance of the issue.

Reviewer 2 Report

I accept the article in the submitted version. The submitted manuscript is well organized, a sufficient number of papers is cited. The Authors created a brief and interesting review concerning TKI in thyroid cancer treatment. The article provides the newest finding in the addressing issue and the potential Readers will benefit from that article.

Author Response

Thank you for the reviewer's comments

Round 2

Reviewer 1 Report

I have no further comments